# Energy-Efficient Clustering and Routing Using ASFO and a Cross-Layer-Based Expedient Routing Protocol for Wireless Sensor Networks

**DOI:** 10.3390/s23052788

**Published:** 2023-03-03

**Authors:** Venkatesan Cherappa, Thamaraimanalan Thangarajan, Sivagama Sundari Meenakshi Sundaram, Fahima Hajjej, Arun Kumar Munusamy, Ramalingam Shanmugam

**Affiliations:** 1Department of Electronics and Communication Engineering, HKBK College of Engineering, Bangalore 560045, Karnataka, India; 2Department of Electronics and Communication Engineering, Sri Eshwar College of Engineering, Coimbatore 641202, Tamilnadu, India; 3Department of Electrical and Electronics Engineering, Amrita College of Engineering and Technology, Erachakulam 629901, Tamilnadu, India; 4Department of Information Systems, College of Computer and Information Sciences, Princess Nourah bint Abdulrahman University, Riyadh 11671, Saudi Arabia; 5Department of Electronics and Communication Engineering, Karpagam Academy of Higher Education, Coimbatore 641021, Tamilnadu, India

**Keywords:** wireless sensor network, sensor nodes, clustering, cross-layer routing, adaptive sailfish optimization

## Abstract

Today’s critical goals in sensor network research are extending the lifetime of wireless sensor networks (WSNs) and lowering power consumption. A WSN necessitates the use of energy-efficient communication networks. Clustering, storage, communication capacity, high configuration complexity, low communication speed, and limited computation are also some of the energy limitations of WSNs. Moreover, cluster head selection remains problematic for WSN energy minimization. Sensor nodes (SNs) are clustered in this work using the Adaptive Sailfish Optimization (ASFO) algorithm with K-medoids. The primary purpose of research is to optimize the selection of cluster heads through energy stabilization, distance reduction, and latency minimization between nodes. Because of these constraints, achieving optimal energy resource utilization is an essential problem in WSNs. An energy-efficient cross-layer-based expedient routing protocol (E-CERP) is used to determine the shortest route, dynamically minimizing network overhead. The proposed method is used to evaluate the packet delivery ratio (PDR), packet delay, throughput, power consumption, network lifetime, packet loss rate, and error estimation, and the results were superior to existing methods. PDR (100%), packet delay (0.05 s), throughput (0.99 Mbps), power consumption (1.97 mJ), network lifespan (5908 rounds), and PLR (0.5%) for 100 nodes are the performance results for quality-of-service parameters.

## 1. Introduction

A wireless sensor network (WSN) contains affordable, tiny sensor nodes. WSNs have proven to be one of the most suitable strategies for transferring data from remote locations to a central data processing station. These sensors self-organize and form a multi-hop network capable of adapting and transmitting compressed data to a base station [1]. Multimedia WSNs support transmitting multimedia data, such as images or videos, and can collect more data in military monitoring, agricultural and industrial monitoring, affordable healthcare, and intelligent buildings. Multimedia data is typically more extensive, and larger sensor nodes can use more resources. Some researchers have attempted to investigate and solve this issue [2]. Energy conservation is one of the most significant aspects for sensor nodes in wireless sensor networks to extend their lifespan. Sending and receiving packets consume the majority of the energy.

Sensor nodes in WSNs frequently use batteries. Because of the network of devices, charging the battery is complex, and the battery’s capacity emerges as the most precious resource for WSNs. As a result, energy conservation becomes a critical issue in WSNs. A new optimization algorithm must be developed to maximize energy efficiency and network lifespan. Clustering is one of the power management functions of WSNs, which splits the network into multiple clusters, with one node in each cluster designated as the cluster head (CH) [3]. By combining the data received from each node and sending it to the base station (BS), the CH reduces the BS’s overhead. Because the BS accepts data from fewer nodes, the WSN saves power in resource-constrained situations. Clustering algorithms aid in reducing power consumption in WSNs [4].

The clustering algorithm works in rounds, each consisting of two phases: formation and stabilization. Nodes are organized into distinct collections, or clusters. Each group has a designated CH. The sensed data is transmitted to the receiver by the CH, which collects data from sensor nodes. In the form of important information, several data aggregation algorithms are employed to cluster algorithms, gather this data, and send it to the BS. A CH is critical for maintaining the energy savings of clustering techniques. The K-Means algorithm chooses a specific number of clusters (*k*), signified by their centroids, and computes the Euclidean distance between any node and all clusters [5,6].

The K-medoids algorithm was developed to enhance the selection of the cluster’s primary nodes and set up a more appropriate network architecture to minimize the sensitivity of outliers [7]. A hybrid energy-efficient distributed clustering algorithm (HEED) is suggested to increase network lifetime. The primary CH nodes in HEED are chosen based on their available power probability, and the sensor nodes (SNs) are selected based on their actual power clusters [8,9]. Efficient data transfer is based on the elephant herding optimization algorithm. This work reduces latency while increasing throughput. Total power consumption is higher; as a result, the network’s life may be shortened. The main limitations of previous systems were performance, network lifetime, latency, packet delivery rate (PDR), power efficiency, and packet loss rate. Furthermore, the limited communication and computing power of wireless sensor networks adds to the complexity of data collection [10].

The primary contribution made by the suggested work is to conserve energy and achieve the maximum lifetime of WSNs. The K-medoids approach is used for clustering nodes with similar data. Each cluster node has a CH chosen using the ASFO algorithm. Furthermore, computation time and CH energy consumption have been minimized because of the shorter transmission range.

## 2. Related Works

A hybrid cross-layer routing protocol was inspired by WSN-assisted energy conservation in the Internet of Things (IoT). Routing is processed at three layers in the cross-layer routing strategy: the physical layer, the data link layer, and the transport layer. The protocol for Bionic Cross-Layer Routing (BiHCLR) is proposed in this paper for efficient and energy-efficient routing in WSN-IoT. Then, to conserve energy in BiHCLR, a fuzzy logic method is used to select a CH for each grid cell [11]. The routing path is then chosen using a hybrid bionic algorithm. The proposed BiHCLR’s performance is assessed using a quality of service (QoS) analysis. Latency-aware heterogeneous cluster-based data acquisition (DA-HCDA) in the IoT was proposed [12]. A DA-HCDA algorithm is presented to ensure maximum coverage. As a result of the introduced DA-HCDA and quartile aggregation mechanisms, end-to-end latency and network lifetime are improved.

A new Energy-Aware Adaptive Fuzzy Neural Clustering (EAANFC-MR) algorithm was implemented. The EAANFC-based clustering method chooses CHs based on residual energy (RE), node distance, and degree [13]. The QOBFO algorithm is used as a multihop routing technique to select the best route to the destination. The proposed EAANFC-MR algorithm has been simulated using MATLAB. Numerous tests demonstrate the advantages of the suggested algorithm when it is examined with various QoS parameters. The Multi-Objective Fuzzy Inference System (moFIS) and Bacterial Feeding Optimization (BFO) are effective algorithms for determining attributes such as degree difference, overall distance to neighbors, remaining energy, and distance to the BS for wireless sensor networks. The demonstrated outcomes revealed that the moFIS-BFO algorithm outperforms and extends network life.

A new Natural Inspired Cross-Layer Clustering (NICC) protocol is investigated along with Bacterial Feeding Optimization (BFO) with an optimum fitness function that simulates the trade-off between data throughput and energy usage. A BFO algorithm selects optimal sensor nodes for routing and clustering issues based on cross-layer parameter fitness value calculations. In various WSN scenarios, demonstrated results revealed that the NICC protocol outperformed state-of-the-art clustering techniques [14]. The Naive Bayes, KNN, and Support Vector Machine (SVM) classifiers are used for classification and image processing applications. The conventional algorithm has less accuracy when compared with the proposed methodology [15].

A new algorithm is proposed for determining the best CH in an IoT-WSN. The Improved Sunflower Optimization (ISFO) algorithm is the name given to this novel algorithm. The proposed algorithm’s results show that it consumes less power than other algorithms and has a higher number of still-alive nodes than others. As a result, the ISFO algorithm demonstrates its superiority in terms of power consumption and network lifespan [16]. An optimal cross-layer-based CH selection is recommended to resolve the energy conservation issue in WSNs. The suggested algorithm with novel probabilistic decision rules is used as a fitness function to find the best path for data transfer. Compared to other methods, CH-IoT minimized power consumption, end-to-end latency, and communication overhead while optimizing network performance. Various performance parameters, such as the number of active nodes, temperature, load, remaining power, and cost function, have been used to select the best CH in an IoT network cluster. The proposed method is compared with several conventional optimization algorithms, including the artificial bee colony, the whale optimization algorithm (WOA), the genetic algorithm, and the adaptive gravity search algorithm [17].

## 3. System Model

WSNs constantly monitor the physical conditions of their surroundings. The WSN infrastructure is composed of a base station (BS) and several SNs.

### 3.1. Network Model

WSN clustering aims to minimize energy usage by dividing sensors into clusters. Common nodes frequently monitor the environment and transmit sensory data to the cluster head. The CH node is always chosen among the common nodes. The significant role of the CH is to accumulate data from each cluster node and send it to the BS. The process of grouping aids in avoiding direct communication between receivers and sensors. The WSN system model is depicted in Figure 1.

### 3.2. Energy Model

The energy of a node is directly proportional to propagation distance ‘*n*’ when the threshold distance ($n_{0}$) exceeds the value of ‘*n*’. The following expression represents the total energy used by each node to transmit the ‘*M*’-bit data packet.
(1)$$E_{tx}\left(M,n\right)=E_{elec}\times M+\varepsilon_{am}\times M$$
(2)$$\varepsilon_{am}=\left\{\begin{array}{c} \varepsilon_{f}\times n^{2},\;\;\;when\;n\leq n_{0}\\ \varepsilon_{p}\times n,\;\;\;when\;n>n_{0} \end{array}\right.$$
where $E_{tx}$ is the total energy required for data transfer, $E_{elec}$ is the dissipation of energy per bit, $\varepsilon_{f}$ is the energy used for amplification in the free space model, *M* is the number of bits, and $n_{0}$ is the threshold transmission distance.

The receiver’s energy consumption is given by
(3)$$E_{rx}\left(M\right)=E_{elec}\times M$$
(4)$$E_{sum}=E_{rx}+E_{tx}$$
where $E_{rx}$ is the total energy required for received data and $E_{sum\;}$ represents the total energy loss for a WSN.

### 3.3. K-Medoids with an Adaptive Sailfish Optimization (ASFO) Algorithm for CH Selection

The sensor nodes are organized into clusters using the clustering technique, and a CH is chosen for each cluster in the wireless sensor network. The primary responsibility of the CH is to gather data from specific cluster nodes and send it to the BS. This research suggests a K-medoids algorithm for clustering in WSNs. The K-medoids algorithm splits all sensor nodes into k clusters. Each cluster is associated with a single object in the K-medoids method. The identified object is called a medoid and corresponds to the cluster’s most central point. The K-medoids group is the shortest distance between clusters because the K-medoids related to the cluster node find the optimal center. It improves communication between sensor nodes, reduces energy consumption, and detects more accurate cluster centers, resulting in shorter packet delays. The K-medoids algorithm is efficient and has a fixed number of convergence steps [18]. The K-medoids clustering algorithm has the following steps:

Step 1: Randomly select *k* points from the input data (where *k* represents the number of clusters to form). 

Step 2: Each data point is assigned to the cluster that contains the closest center point.

Step 3: For each data point in cluster ‘*i*’, calculate and add the distance from all other data points. Specify the point of the i cluster that reduces the total calculated distance from other points as the center point of the cluster.

Step 4: Repeat steps 1 and 3 until convergence is reached, that is, until the center point stops moving.

The clustering method involves grouping sensor nodes and selecting CHs for all groups in the WSN. The CH’s primary function is to collect information from a specific node in the cluster and transmit it to the BS. Because of its ability to find exact central clusters, or centroids, cluster formation is accomplished using the K-medoids algorithm, resulting in low power consumption, low packet latency, and better sensor nodes. The process involves estimating the number of clusters and calculating the first CH node using Equation (5):(5)$$c=\sqrt{\frac{n}{2}}$$
where *n* denotes the number of nodes. These algorithms determine the initial mean point and center location (*L*) for all the nodes derived.
(6)$$L=\frac{\sum_{n=1}^{N}x_{n}}{n}$$
where $x_{n}$ represents the coordinate of the sensor. The average separation between the *SN* and *L* is represented as *D* in the following formula:(7)$$D=\frac{\sum_{n=1}^{N}\left|x_{n}-L\right|}{n}$$

The separation between the sensor nodes *(SN)* and center location *(L)* can be used to calculate the centroid, and clustering is obtained using Equation (7) until a proper CH is selected. The procedure for formation of clustering using K-medoids with a Sailfish Optimizer is described in Algorithm 1.

**Algorithm 1:** Algorithm for clustering using K-medoids with a Sailfish Optimizer**Network Initialization**    Step 1: Initialization of the WSN    Step 2: Locate BS at coordinates (50, 180)    Step 3: Place all the SNs arbitrarily**Formation of clusters using K-medoids and selection of CH using ASFO**    Step 4: Number of nodes *N* divided into several clusters    Step 5: Every cluster has *N* nodes, and each node is related to its nearest CH    Step 6: Randomly select the first CH by selecting the first random medoid from *N* in the cluster    Step 7: Three-dimensional coordinates (x, y, z) are generated by every normal node to CH    Step 8: K-means distance calculation is performed by the CH    Step 9: ASFO algorithm is used to select the new CH and center the cluster node    Step 10: Repeat step 7 to 9 until the node in the absolute center is found**End**

The specific coordinate values used in the algorithm, such as (50, 180), have significance beyond being randomly generated coordinates within the range of 0 to 200. As a result, the best solution was obtained by employing the ASFO algorithm process. The CH is chosen from the solution, and then it uses a cross-layered expedient routing protocol to find the best traversal path to send the collected data from the transmitter to the receiver.

### 3.4. E-CERP Routing Algorithm for a WSN

The energy-efficient cross-layer-based expedient routing protocol (E-CERP) transfers data over the shortest paths in an energy-efficient and scalable manner. It increases communication link reliability while reducing packet delays. The existing cross-layer-based opportunistic routing protocol (CORP) algorithm has several drawbacks:High data transmission complexity occurs when the number of constraints increases due to the limited computing power of the WSNs.They are challenging to integrate.They have a high power consumption, packet loss, delivery rate, and communication delay.

The proposed E-CERP technology solves the problems of conventional routing protocols, and the goal is to maximize the nodes’ transmission power by utilizing the network’s remaining energy.

(i) Local broadcast

Each node sends a HELLO message that includes its ID, the number of hops to the BS, and the path cost. The base station path cost can change from round to round if it is initially 0 and each node is infinite. Each node organizes its list of neighbors based on the HELLO messages it receives. Signal strength is calculated using the received signal strength indicator (RSSI) of incoming messages. The average RSSI calculates the link reliability metric *L*(*n*, *m*), expressed in Equation (8).
(8)$$\;\;L_{n,m}=T_{mtp}-T_{arp}$$

The Lagrange multiplier is used to estimate the link cost as follows:(9)$$cos\left(n,\;m\right)=X_{cir}+X_{arp}+\frac{In\;I}{v}.\frac{1}{h_{nm}}$$

(ii) Routing algorithm

A parent *Par*(*n*) has to be chosen for a specified sensor node *n* to act as the next hop sensor node to send data from node *n* to the BS. The mathematical equation can be written as follows:(10)$$Par\left(n\right)=arg_{m\in N_{i}\left(n\right)}\;min\left[cost\left(m\right)+cost\left(n,m\right)\right]$$

Using the cost function and the parent route selection shown in Equation (10), a cost-based route is created. In each round, the parent preferences are updated.

(iii) Transmission Power Control (TPC)

The ideal transmit power can be determined using hop count, as represented in Equation (11):(11)$$T_{tx\left(n\right)}=\frac{1}{h_{n}}\left[ln\;I+ln\;\left(h_{n}\sum_{n=1}^{H}\frac{1}{h_{n}}\right)\right]$$

For the transmitted execution, reasonable simplification is carried out, and the upper bound can be calculated using Equation (12):(12)$$\left(\sum_{n=1}^{H}\frac{1}{h_{n}}\right)ln\;I\leq \sum_{n=1}^{H}X_{tx\left(n\right)}\leq \left(\sum_{n=1}^{H}\frac{1}{h_{n}}\right)\left(ln\;I+ln\;H\right)$$

Thus, for the given sensor node, the upper bound is estimated using Equation (13).
(13)$$X_{tx}\left(n\right)=\frac{1}{h_{nm}}\;\left[ln\;I+ln\;H\left(n\right)\right]$$
where “*I*” is the target end-to-end success probability and “*H*” represents the hop count. The ideal path for sending and receiving data has been determined after evaluating the transmitting power. It has negligible packet loss and delays, as well as minimal energy conservation.

## 4. Experimental Results and Discussion

The simulation is based on data transfer, node availability, and sensing range. The dataset used in this work is the energy efficiency detection dataset. In one dataset, there are a total of 780 data samples, which are used to conduct experiments on the proposed technique. The parameters considered for simulation setup are given in Table 1.

### 4.1. Performance Analysis of Clustering and Routing

There are multiple sensor nodes in a WSN that can be alive or dead. The proposed clustering technique identifies distances between nodes by regularly monitoring the nodes and forming clusters based on distances. The K-medoids algorithm is used to group nearby nodes into clusters, and the selection of the cluster head is based on the SN’s energy usage. As CH, the energy-efficient SN is chosen, which can be changed anytime. Figure 2 depicts the transmission of data from a sensor node to the cluster head. According to the diagram, six cluster heads manage all the SNs in a cluster.

Compared to existing methods, the K-medoids algorithm and E-CERP suffer from high packet loss, delivery delays, and a lack of stable power when using existing processes to transmit data. The proposed K-medoids clustering approach is compared with the K-means [18] and Fuzzy C-means approaches, which include particle swarm optimization and krill herd optimization algorithms [19,20,21,22]. The introduced ASFO method is compared with two existing optimization approaches: the krill swarm optimization algorithm [23] and the whale optimization algorithm (WOA) [24]. The proposed results were compared with cross-layer-based adaptive thresholding (CLAT) and cross-layer fuzzy logic (CLFL).

### 4.2. Performance Analysis of Clustering

Clustering is an essential method for ensuring efficient data transmission. The proposed work used the K-medoids approach to cluster similar data into one group based on the average energy consumption of each node [25,26].

Figure 3 and Table 2 represent the comparison graph for the average energy consumption by the proposed approach and the existing methods, K-means and Fuzzy C-means clustering. Therefore, the hybrid approach consumed 0.0196 J on average for cluster size 2, whereas the K-means method consumed 0.0245 J and the Fuzzy C-means logic approach consumed 0.0294 J.

Table 3 and Figure 4 show the comparison of CH selection by the proposed ASFO method with two conventional methods, the krill herd algorithm and the WOA. The CH selection in 250 rounds by the proposed method based on the average residual energy is 39.2337 J, whereas the existing methods, the krill herd algorithm and the WOA, have 34.3295 J and 29.4253 J, respectively.

Figure 4 and Figure 5 show that the proposed ASFO algorithm is more effective than the conventional algorithm for clustering and CH selection. Thus, the proposed method efficiently formed the cluster and chose the CH for the corresponding cluster.

### 4.3. Energy Consumption

The energy consumption of the introduced algorithm is shown in Figure 5, along with two conventional algorithms: a cross-layer-based adaptive threshold technique and a cross-layer fuzzy logic approach.

From the experimental results, the proposed CERP consumed less energy (1.97 mJ), whereas the existing cross-layer-based adaptive threshold technique consumed 7.75 mJ and the cross-layer fuzzy logic approach consumed 8.43 mJ. For 200 nodes, the E-CERP approach consumed 1.10 mJ of energy, and the existing techniques, cross-layer-based adaptive threshold and cross-layer fuzzy logic, consumed 5.51 mJ and 10.36 mJ of energy, respectively. For 300 nodes, the proposed technique consumed 5.13 mJ of energy, the cross-layer-based adaptive threshold technique consumed 7.91 mJ of energy, and the cross-layer fuzzy logic approach consumed 9.53 mJ. For 400 nodes, the introduced E-CERP approach consumed 4.8 mJ, whereas the cross-layer-based adaptive threshold technique and cross-layer fuzzy logic approach consumed 7.45 mJ and 11.23 mJ, respectively. For 500 nodes, the introduced approach consumed 4.19 mJ of energy, whereas the existing cross-layer-based adaptive threshold technique and the cross-layer fuzzy logic approach consumed 7.55 mJ and 10.37 mJ of energy, respectively. So, the proposed method is shown to be very effective in data transmission.

### 4.4. Network Lifetime

The proposed CERP’s network lifetime is compared with the conventional algorithms in Figure 6. This method attained a higher network lifetime than the remaining methods. Table 4 shows the network lifespan and energy consumption of the proposed E-CERP and conventional approaches.

From the simulation results, it is evident that the network lifetime is extended by the proposed method for 100 nodes (5908 rounds). In contrast, the lifetimes of an existing cross-layer-based adaptive threshold technique and cross-layer fuzzy logic approaches are 5395 and 4904 rounds, respectively. Moreover, as the number of nodes increases, the network lifetime decreases. Hence, data transmission efficiency is based on the increased network lifetime.

### 4.5. Throughput

The suggested technique is compared with a cross-layer-based adaptive threshold and cross-layer fuzzy logic techniques in Table 5.

Figure 7 and Figure 8, and Table 5, show the evaluation of the proposed technique and the conventional algorithms with respect to end-to-end delay and throughput. The E-CERP approach’s throughput was high (0.99 Mbps) in 100 nodes compared to the conventional algorithms.

From Table 5, it is seen that the number of nodes increases as the throughput decreases. The throughputs obtained by the conventional cross-layer-based adaptive threshold technique and cross-layer fuzzy logic approach are 0.99 Mbps and 0.96 Mbps, respectively, for 100 nodes.

### 4.6. End-to-End Delay

The comparison of the end-to-end delays of the proposed CERP and existing approaches is represented in Figure 9 and Table 5. This method attained a lower delay in 100 nodes (0.05 s), and the time delay increased as the number of nodes increased in the WSN. For 100 nodes, the time delays of the two existing methods, the cross-layer-based adaptive threshold technique and cross-layer fuzzy logic, are 1.97 s and 2.94 s, respectively.

### 4.7. Packet Delivery Ratio (PDR)

Table 6 compares packet delivery and packet loss ratios to the number of nodes. Figure 9 and Figure 10 show that the PDR decreased when the number of nodes increased. The PDRs obtained using different algorithms for 100 nodes were 97% for the cross-layer-based adaptive threshold technique and 95% for the cross-layer fuzzy logic method. If the PDR value is high, all data from the base station will be received without any data loss. Therefore, the proposed approach achieves high performance efficiency when compared to other algorithms.

### 4.8. Packet Loss Ratio (PLR)

The packet loss ratio (PLR) of the proposed method and other existing methods are shown in Figure 10. For 100 nodes, the proposed CERP obtained 0% PLR (i.e., there was no loss), and the existing methods, cross-layer-based adaptive threshold and cross-layer fuzzy logic approaches, achieved high PLR values of 2.58% and 4.54%, respectively. It shows that the proposed CERP is more effective for data transmission. The PLR increases when the number of nodes increases. The graph indicated that the introduced method significantly decreased packet loss compared to other energy-efficient algorithms.

### 4.9. Jitter

The comparison of jitter for the proposed technique and existing approaches is shown in Figure 11. The E-CERP algorithm obtained a low jitter value of 0.15 ms for 100 nodes. In contrast, the conventional algorithms, cross-layer-based adaptive threshold and cross-layer fuzzy logic approach, attained 0.25 ms and 0.39 ms, respectively. For 200 nodes, the jitter value of the proposed method was 0.12 ms; however, the conventional algorithms attained 0.20 ms and 0.34 ms, respectively.

For 300 nodes, the jitter value obtained by the E-CERP algorithm was 0.10 ms, and the conventional algorithms attained 0.17 ms and 0.24 ms, respectively. For 500 nodes, the jitter value of the suggested approach was 0.05 ms. However, the existing approaches, cross-layer-based adaptive threshold and cross-layer fuzzy logic technique, attained 0.10 ms and 0.15 ms, respectively. As a result, by evaluating the performance of cross-layer-based adaptive threshold, cross-layer fuzzy logic, K-means, and Fuzzy C-means approaches, the introduced E-CERP algorithm performances are resolved based on the analysis of QoS parameters. Based on the simulation results, the power consumed by the network is low and the network lifetime is high for the proposed method.

### 4.10. Evaluation of the Proposed ASFO and E-CERP Approach with Conventional Techniques

The particle swarm optimization is utilized for the optimization problem based on fuzzy clustering. Still, it has some disadvantages, such as that it is easy to fall into local optima and the convergence rate is slow, while the HHO algorithm is simple, flexible, easy to implement, and has a high convergence rate. In fuzzy and artificial bee colony-based implementations of MAC, the clustering, routing, and data delivery (FABC-MACRD)-based cross-layered method was used for the clustering and data transmission, in which the energy consumption was high, including the energy consumed for 500 nodes. In contrast, the proposed CERP method consumed less energy (4.19 J), and the packet delivery ratio of FABC-MACRD was 79%, while the proposed CERP was 96%. Compared to FABC-MACRD, the proposed method extended the network’s lifespan, and the proposed K-medoids with ASFO and E-CERP improve the reliability of the communication link and reduce packet delay.

## 5. Conclusions

This research develops cluster head selection using K-medoids with ASFO and clustering and multihop routing protocol (CMRP) algorithms for efficient routing in WSNs. For optimal CH selection from suitable nodes, the K-ASFO approach can be used. The outcomes were compared using three standard metrics: throughput, residual power, and first dead node. Finally, the CMRP algorithm routing protocol is used for efficient data transfer. The routing process is permitted on recognized node-to-node paths. The simulation results demonstrate that the proposed system outperforms other existing algorithms. The performance of the proposed methodology is compared to that of existing optimization-based routing algorithms. CMRP chooses the best path from the cluster head to the sink node. The proposed method is used to evaluate PDR, packet delay, throughput, power consumption, network lifetime, packet loss rate, and error estimation, and the results were superior to existing methods. PDR (100%), packet delay (0.05 s), throughput (0.99 Mbps), power consumption (1.97 mJ), network lifespan (5908 rounds), and PLR (0.5%) for 100 nodes are the performance results for QoS parameters. The proposed method has an overall accuracy of 93.19%. As a result, the proposed approach outperforms existing strategies in terms of overall performance. The proposed method produced better results in all scenarios and metrics. The proposed work was idealized, implemented, and tested against a static sensor node WSN. This research can be expanded to mobile sensor nodes or networks with sensors that can change position in real-time.

## Figures and Tables

**Figure 1 sensors-23-02788-f001:**
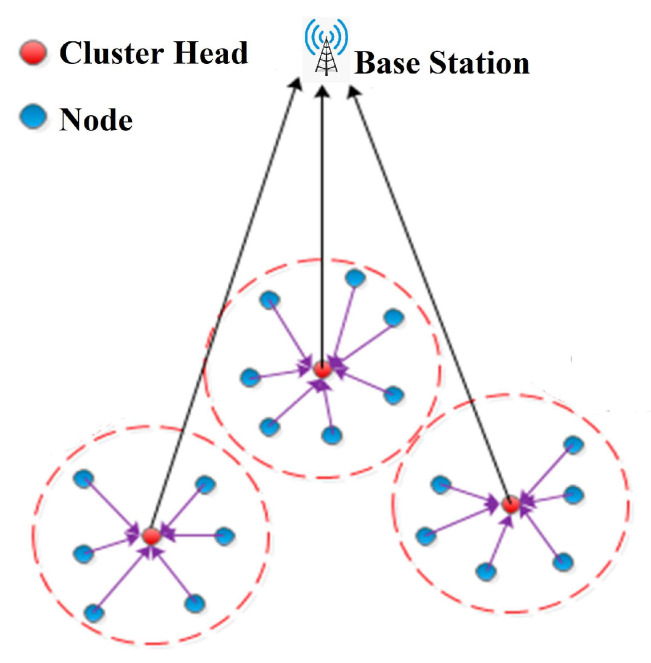
General architecture of a wireless sensor network.

**Figure 2 sensors-23-02788-f002:**
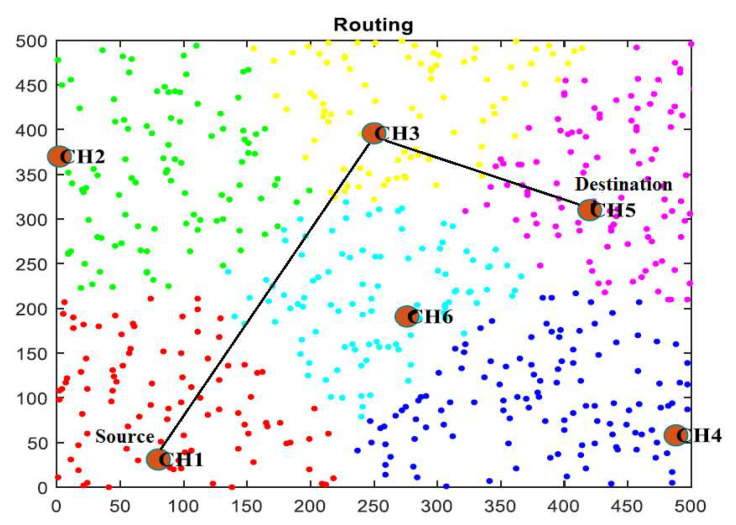
Data path from the source node CH1 to the destination node CH5.

**Figure 3 sensors-23-02788-f003:**
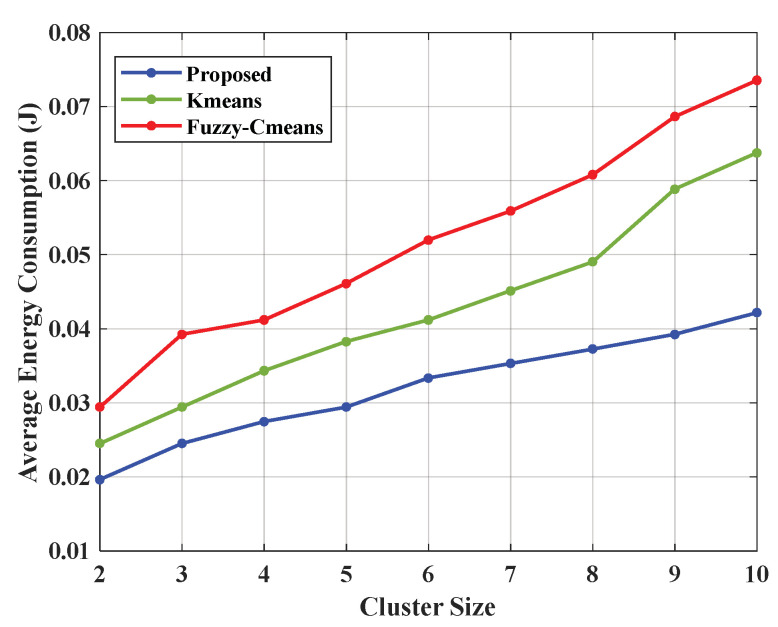
Performance analysis of the energy consumption of the proposed ASFO algorithm.

**Figure 4 sensors-23-02788-f004:**
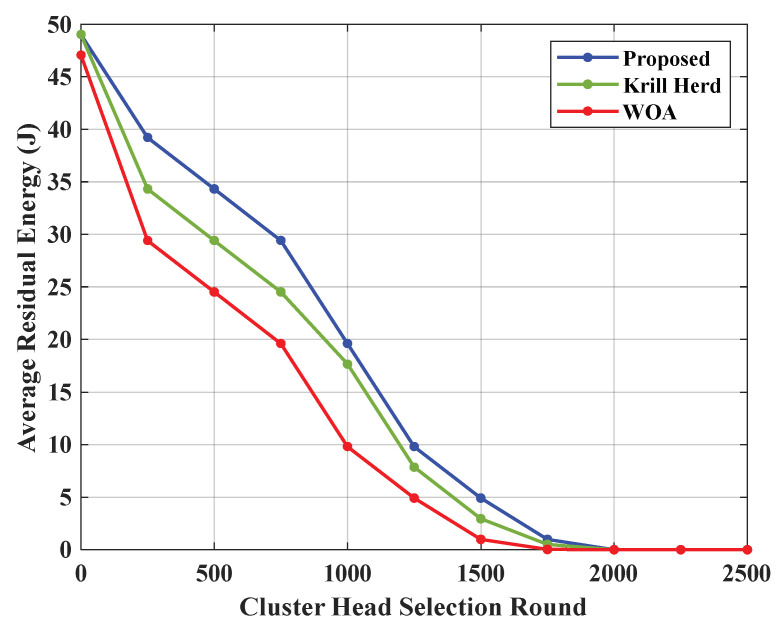
Comparison of CH selection by the proposed method with existing approaches.

**Figure 5 sensors-23-02788-f005:**
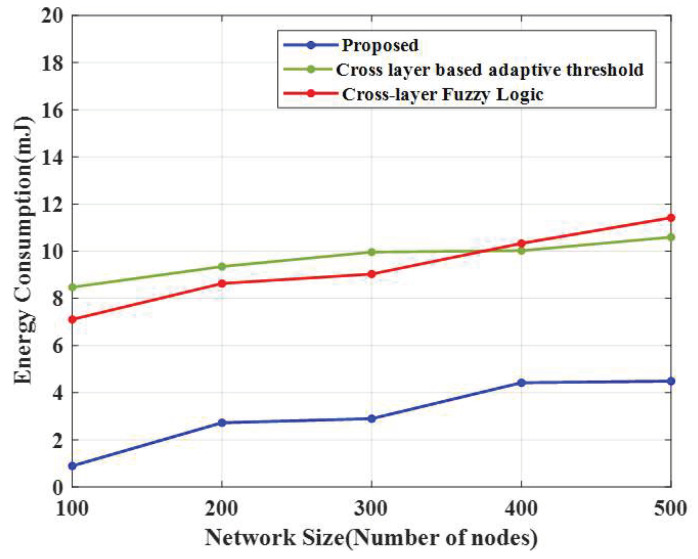
Energy consumption comparison.

**Figure 6 sensors-23-02788-f006:**
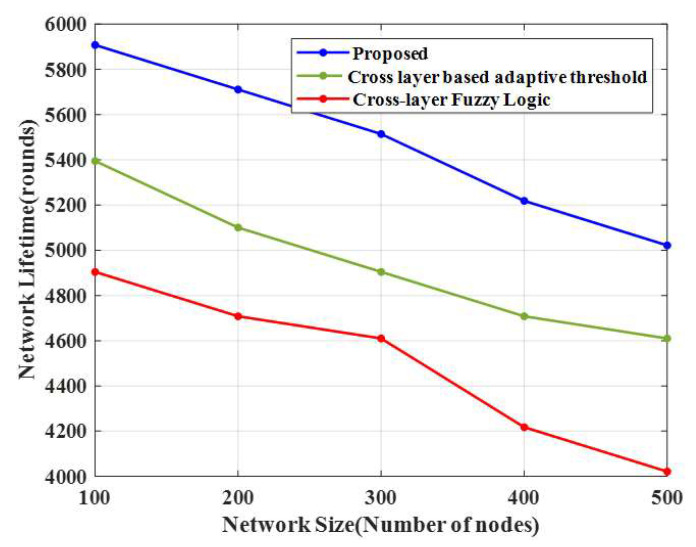
Comparison of the network lifetime.

**Figure 7 sensors-23-02788-f007:**
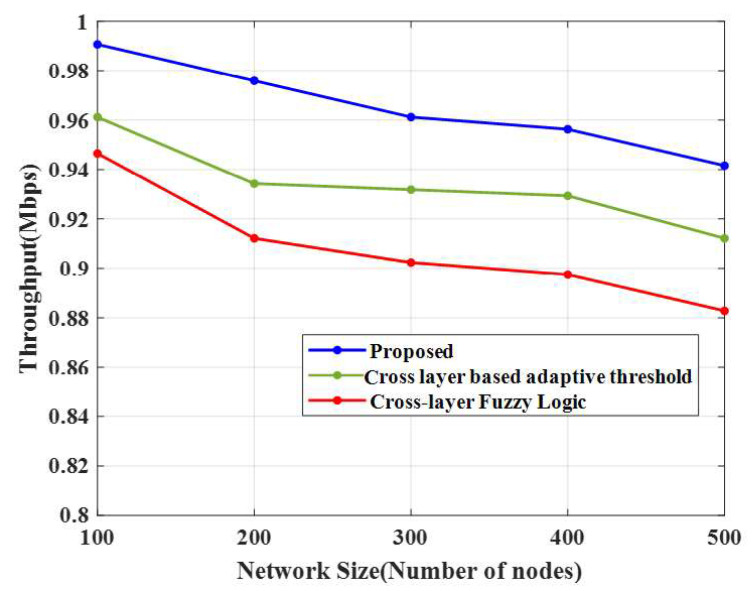
Throughput comparison.

**Figure 8 sensors-23-02788-f008:**
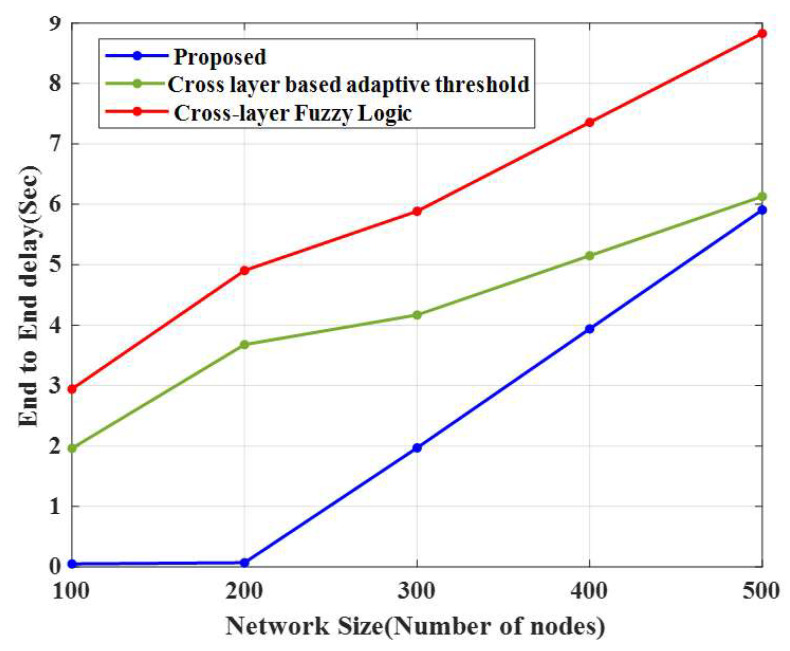
Performance of end-to-end delay.

**Figure 9 sensors-23-02788-f009:**
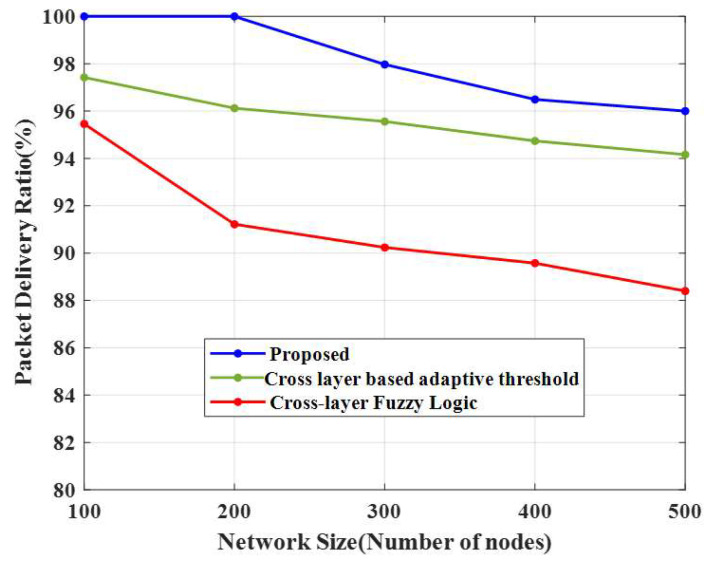
Comparison of the packet delivery ratio.

**Figure 10 sensors-23-02788-f010:**
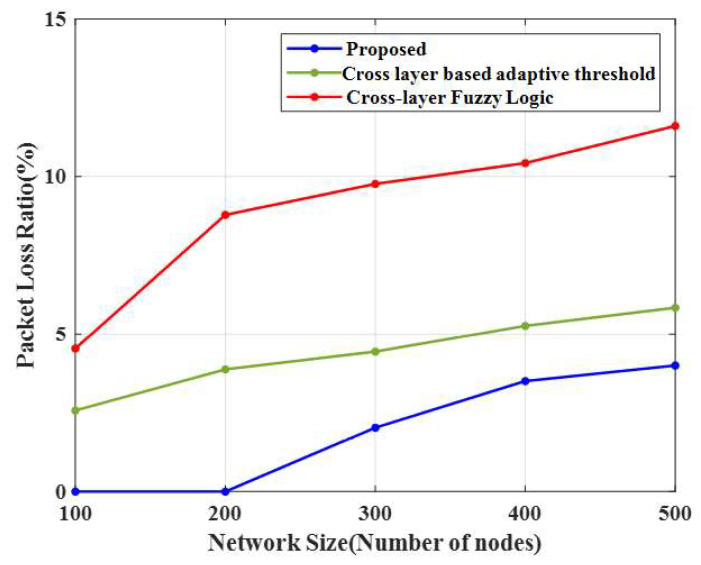
Comparison of the packet loss ratio.

**Figure 11 sensors-23-02788-f011:**
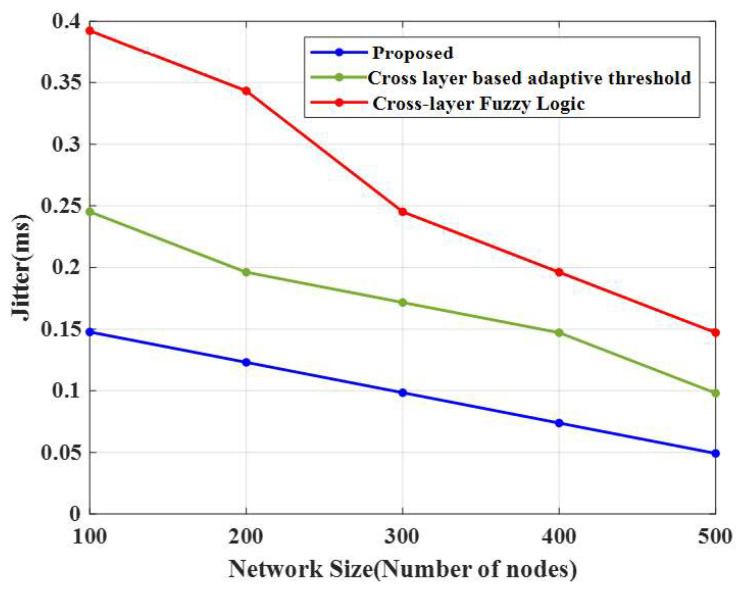
Comparison of jitter.

**Table 1 sensors-23-02788-t001:** Parameters for simulation setup.

Parameters	Value
Number of nodes	500
Deployment area	500 × 500
Total Clusters	6
Packet size	512 bytes
Packet sending rate	1 packet/s
Initial energy	0.5 J
Data samples	55

**Table 2 sensors-23-02788-t002:** Comparison of the average energy consumption of the proposed ASFO algorithm.

Approaches	Average Energy Consumed (J) vs. Cluster Size								
	2	3	4	5	6	7	8	9	10
K-means	0.024	0.029	0.0343	0.038	0.041	0.045	0.049	0.059	0.0638
Fuzzy C-means	0.029	0.039	0.041	0.046	0.052	0.056	0.061	0.069	0.074
Proposed	0.019	0.023	0.028	0.029	0.033	0.035	0.037	0.039	0.0423

**Table 3 sensors-23-02788-t003:** Average residual energy consumption of the proposed ASFO algorithm.

Approaches	Average Energy Consumed (J) vs. Rounds										
	0	250	500	750	1000	1250	1500	1750	2000	2250	2500
Krill herd	49.04	34.33	29.43	24.52	17.66	7.85	2.94	0.49	0	0	0
WOA	47.08	29.43	24.52	19.62	9.81	4.9	0.98	0.02	0	0	0
Proposed	49.04	39.23	34.33	29.43	19.62	9.81	4.9	0.98	0	0	0

**Table 4 sensors-23-02788-t004:** Energy consumption and network lifespan.

Techniques	Average Energy Consumed					Network Lifespan				
Number of Nodes	100	200	300	400	500	100	200	300	400	500
Proposed CERP	1.97	1.10	5.13	4.80	4.19	5908	5711	5514	5218	5022
CLAT	7.75	5.51	7.91	7.45	7.55	5395	5100	4904	4708	4610
CLFL	8.43	10.36	9.53	11.23	10.37	4904	4708	4610	4218	4021

**Table 5 sensors-23-02788-t005:** Throughput and the end-to-end delay comparison.

Methods	Throughput					End-to-End Delay				
Number of Nodes	100	200	300	400	500	100	200	300	400	500
Proposed CERP	0.99	0.98	0.96	0.95	0.94	0.0492	0.0689	1.9692	3.9384	5.9076
CLAT	0.96	0.93	0.93	0.93	0.91	1.9617	3.6785	4.1686	5.1494	6.1302
CLFL	0.95	0.91	0.90	0.90	0.88	2.9425	4.9042	5.8851	7.3563	8.8276

**Table 6 sensors-23-02788-t006:** Comparison of PDR and PLR.

Techniques	PDR					PLR				
Number of Nodes	100	200	300	400	500	100	200	300	400	500
Proposed CERP	100	100	98	96	96	0	0	2.03	3.51	4
CLAT	97	96	96	95	94	2.57	3.87	4.44	5.26	5.83
CLFL	95	91	90	90	88	4.53	8.78	9.76	10.42	11.60

## Data Availability

The study did not report any data.

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
