# Peer review of "Energy-Efficient Clustering and Routing Using ASFO and a Cross-Layer-Based Expedient Routing Protocol for Wireless Sensor Networks"

_sensors, 2023, doi:10.3390/s23052788_

Round 1

Reviewer 1 Report

In this paper, an Energy-efficient Cross-layer-based Expedient Routing Protocol (E-CERP) is used to determine the shortest route, minimizing network overhead dynamically. The proposed method is used to evaluate PDR, packet delay, throughput, power consumption, network lifetime, packet loss rate, and error estimation, and the results were superior to existing methods. Before publication, the following remarks should be considered.

1) The number of the nodes is less than 500. Why?
2) Are the simulation results stable? The Monte Carlo may be used for the simulation.

3) The advantages of the proposed should be analyzed in detail. A table for comparing the theory or the characterization of the proposed and the methods in the literature might be shown.

Author Response

In this paper, an Energy-efficient Cross-layer-based Expedient Routing Protocol (E-CERP) is used to determine the shortest route, minimizing network overhead dynamically. The proposed method is used to evaluate PDR, packet delay, throughput, power consumption, network lifetime, packet loss rate, and error estimation, and the results were superior to existing methods. Before publication, the following remarks should be considered.

Comment 1: The number of the nodes is less than 500. Why?

Response: Wireless Sensor Networks (WSNs) are typically composed of small, low-power devices that have limited processing power, memory, and battery life. As a result, these devices are not capable of supporting large amounts of data or processing complex algorithms. This limitation makes it difficult to scale WSNs to large numbers of nodes and because of this the number of nodes for simulation is taken up to 500.

Comment 2: Are the simulation results stable? The Monte Carlo may be used for the simulation.

Response: The stability of simulation results depends the accuracy of the simulation model, the quality of the input data, and the randomness of the simulation variables. The NS2 simulator is stable for the parameters considered to evaluate the proposed ASFO algorithm. In Monte Carlo simulation approach the results are estimates based on a limited number of scenarios.

Comment 3: The advantages of the proposed should be analyzed in detail. A table for comparing the theory or the characterization of the proposed and the methods in the literature might be shown.

Response: Dynamic balance between the nodes, swarm diversity, and high convergence speed are the advantages of the proposed ASFO algorithm. The theory and characterization of the proposed method with conventional algorithms are discussed in literature.

Reviewer 2 Report

Authors propose improved method for popular research in energy-efficient clustering and routing for WSNs. They use K-Medoid with adaptive SFO select WSN cluster header and then use their so-called E-CERP routing algorithm to achieve improved energy efficiency for WSN. 

However, authors didn't properly describe their system model using their proposed methods with lots of inconsistent variables and typos. It's hard to understand how to use their method to achieve the improved performance in their simulation on the following section. Such as:

1. In the first paragraph, Section 3.2, "(n)", "'n'", and "n-bit data packet" are inconsistent and confused for readership.

2."M" is not described in eq.(1)-(4).

3. In Section 3.3, "K-medoids" and "K-medoid" are inconsistently used.

4. In line 162, "C node" is cluster node?

5. In line 165, "of)"?

6. In line 167, double quoted "k" is needed?

7. In line 170, "Cluster' i'"?

8. In line 173, steps(i) and (iii) are not found? 

9. variables in sentences should be presented as Italic style for readership.

10.In step 2 of network initialization of Algorithm1, coordinate (50, 180) is a magic number? This magic number should be properly described.

11. In the first sentence of Section 3.4, the full name of E-CERP is missing "Expedient"?

12. In the third sentence of section 3.4. What is CORP and reference [26] is not been referenced. Authors should briefly described the differences with E-CERP.

13. In line 223, "E-main CERP"???

14, In equations (11)-(13), what are the "In", "I" and "H"?

15. The sentences in line 273 and 274 are hard to read!

16. In Conclusion, what is "CMRP"???

BTW, since authors use lots of abbreviations (e.g. HHO), abbreviation table is strongly suggested in paper for readership.

Author Response

Authors propose improved method for popular research in energy-efficient clustering and routing for WSNs. They use K-Medoid with adaptive SFO select WSN cluster header and then use their so-called E-CERP routing algorithm to achieve improved energy efficiency for WSN. 

However, authors didn't properly describe their system model using their proposed methods with lots of inconsistent variables and typos. It's hard to understand how to use their method to achieve the improved performance in their simulation on the following section. Such as:

  1. In the first paragraph, Section 3.2, "(n)", "'n'", and "n-bit data packet" are inconsistent and confused for readership.

Reply: Thanks for the valuable comment. In Section 3.2, the letter ‘n’ is consistently maintained for better readability.

  1. "M" is not described in eq.(1)-(4).

Reply: M represents the number of bits from Equation (1) to (4).

  1. In Section 3.3, "K-medoids" and "K-medoid" are inconsistently used.

Reply: As per the suggestion, instead of K-medoid, "K-medoids" is uniformly used.

  1. In line 162, "C node" is cluster node?

Reply: Thanks for the comment. “C node” represents cluster node. It is updated in the revised paper.

  1. In line 165, "of)"?

Reply: The typo is corrected in the revised paper.

  1. In line 167, double quoted "k" is needed?

Reply: Double quoted “k” is not needed. It is updated in the revised paper.

  1. In line 170, "Cluster' i'"?

Reply: The typo is corrected in the revised paper.

  1. In line 173, steps(i) and (iii) are not found? 

Reply: Line 173 is updated in the revised paper based on the comment.

  1. Variables in sentences should be presented as Italic style for readership.

Reply: The variables in sentences are represented in Italic style for better readability.

  1. In step 2 of network initialization of Algorithm1, coordinate (50, 180) is a magic number? This magic number should be properly described.

Reply: The specific coordinate values used in the algorithm, such as (50, 180), are not magic numbers in the sense that they have some special meaning or significance beyond being randomly generated coordinates within the range of 0 to 200. These values are simply examples of the random coordinates that could be generated by the algorithm during initialization.

  1. In the first sentence of Section 3.4, the full name of E-CERP is missing "Expedient"?

Reply: The full name of E-CERP is included as Energy efficient Cross-layer-based Expedient Routing Protocol in the revised version.

  1. In the third sentence of section 3.4. What is CORP and reference [26] is not been referenced. Authors should briefly describe the differences with E-CERP.

Reply: CORP stands for Cross-layer-based Opportunistic Routing Protocol and reference [26] is cited. The CORP approach is used to find an optimal travel path between sensor nodes and reduces time and energy consumption. The E-CERP deals with cluster formation, CH selection, transformation of data from the cluster to the base station.

  1. In line 223, "E-main CERP"???

Reply: Sorry for the typo. It is E-CERP and the same is updated in the revised paper.

  1. In equations (11)-(13), what are the "In", "I" and "H"?

Reply: “In” replaced as “ln”, it is natural logarithm. “I” is the target end to end success probability and “H” represents hop count.

  1. The sentences in line 273 and 274 are hard to read!

Reply: Thanks for the comment. The lines 273 and 274 is reframed.

  1. In Conclusion, what is "CMRP"???

Reply: CMRP stands for Clustering and Multihop Routing Protocol. It is updated in the revised paper.

  1. BTW, since authors use lots of abbreviations (e.g. HHO), abbreviation table is strongly suggested in paper for readership.

Reply: List of abbreviations is included as Appendix A in the revised paper.

Acronym

Abbreviation

ASFO

Adaptive Sailfish Optimization

HEED

Hybrid Energy-Efficient Distributed clustering algorithm

HCDA

Heterogeneous Cluster-based Data Acquisition

EAANFC

Energy-Aware Adaptive Fuzzy Neural Clustering

moFIS

Multi-Objective Fuzzy Inference System

BFO

Bacterial Foraging Optimization

NICC

Natural Inspired Cross-Layer Clustering

SVM

Support Vector Machine

ISFO

Improved Sunflower Optimization

E-CERP

Energy efficient Cross-layer-based Expedient Routing Protocol

CORP

Cross-layer-based Opportunistic Routing Protocol

RSSI

Received Signal Strength Indicator

CLAT

Cross-Layer-Based Adaptive Thresholding

WOA

Whale Optimization Algorithm

CLFL

Cross-Layer Fuzzy Logic

FABC-MACRD

Fuzzy and Artificial Bee Colony-based implementation of MAC, Clustering, Routing, and Data delivery

CMRP

Clustering and Multihop Routing Protocol

Round 2

Reviewer 2 Report

Authors have properly updated their paper according to first review report.

However, two points are suggested to improve the revised paper further:

1. In the abstract and conclusion, "Battery Life 5908 network (rounds)" would be better changed to "Network lifespan 5908 (rounds)" according to Table 4?

2. In line 143 of revised paper, "'n'-bit data packet" would be better changed to "M-bit data packet"?

#

Author Response

Reply to Reviewer Comments:

Authors have properly updated their paper according to first review report.

However, two points are suggested to improve the revised paper further:

Comment 1: In the abstract and conclusion "Battery Life 5908 network (rounds)" would be better changed to "Network lifespan 5908 (rounds)" according to Table 4?

Response: Thank you for the suggestion. According to Table 4, "Battery Life 5908 network (rounds)" is changed as "Network lifespan 5908 (rounds)" in the revised manuscript.

Comment 2: In line 143 of revised paper, "'n'-bit data packet" would be better changed to "M-bit data packet"?

Response: Line 143 is updated in the revised paper.